# A R2R3-MYB Transcription Factor, *VvMYBC2L2*, Functions as a Transcriptional Repressor of Anthocyanin Biosynthesis in Grapevine (*Vitis vinifera* L.)

**DOI:** 10.3390/molecules24010092

**Published:** 2018-12-27

**Authors:** Ziguo Zhu, Guirong Li, Li Liu, Qingtian Zhang, Zhen Han, Xuesen Chen, Bo Li

**Affiliations:** 1Shandong Institute of Pomology, Shandong Academy of Agricultural Science, No 66 Longtan Road, Taian 271000, Shandong, China; shanhong98@163.com (Z.Z.); 15153871569@163.com (L.L.); zhangqingtian@shandong.cn (Q.Z.); gsshanzhen@shandong.cn (Z.H.); 2College of Horticulture and Landscape Architecture, Henan Institute of Science and Technology, Xinxiang 453003, Henan, China; liguirong10@163.com; 3College of Horticulture Science and Engineering, Shandong Agriculture University, No 61 Daizong Road, Taian 271000, Shandong, China; chenxs@sdau.edu.cn

**Keywords:** grapevine, anthocyanin, MYB transcription factor, repressor, flavonoid pathway

## Abstract

In grapevine, the MYB transcription factors play an important role in the flavonoid pathway. Here, a R2R3-MYB transcription factor, *VvMYBC2L2*, isolated from *Vitis vinifera* cultivar Yatomi Rose, may be involved in anthocyanin biosynthesis as a transcriptional repressor. *VvMYBC2L2* was shown to be a nuclear protein. The gene was shown to be strongly expressed in root, flower and seed tissue, but weakly expressed during the fruit development in grapevine. Overexpressing the *VvMYBC2L2* gene in tobacco resulted in a very marked decrease in petal anthocyanin concentration. Expression analysis of flavonoid biosynthesis structural genes revealed that chalcone synthase (CHS), dihydroflavonol 4-reductase (DFR), leucoanthocyanidin reductase (LAR) and UDP glucose flavonoid 3-*O*-glucosyl transferase (UFGT) were strongly down-regulated in the *VvMYBC2L2*-overexpressed tobacco. In addition, transcription of the regulatory genes AN1a and AN1b was completely suppressed in transgenic plants. These results suggested that *VvMYBC2L2* plays a role as a negative regulator of anthocyanin biosynthesis.

## 1. Introduction

Flavonoids (including flavonols, anthocyanins and proanthocyanidins) are important plant secondary metabolites, which play multi-biological roles in plants [1,2]. In grapevine, fruit skin color is mainly determined by the content and composition of anthocyanins which also contribute to wine organoleptic properties such as color, taste, bitterness and astringency [3,4]. As a consequence, it is valuable to understand the regulatory mechanism of flavonoid biosynthesis in grapevine.

The MYB family is one of the largest transcription factor families in higher plants, and it plays an important role in the flavonoid pathway. Based on the number of the highly repeat conserved domains (R), MYB proteins can be classified into four major types: 2R-MYB (R2R3-MYB), 3R-MYB (R1R2R3-MYB), 4R-MYB (R1R2R2R1), and MYB- related proteins (or 1R-MYB) [5,6]. Among these MYB proteins, many R2R3-MYB proteins have been shown to be involved in the regulation of proanthocyanin or anthocyanin biosynthesis as positive regulators. In apple, *MdMYB1*, *MdMYB3*, *MdMYB9*, *MdMYB10* and *MdMYB11* activated the expression of the anthocyanin biosynthesis structural genes [7,8,9,10]. In *Arabidopsis*, *PRODUCTION ANTHOCYANIN PIGMENT1* (*PAP1*) and *PAP2* induced the accumulation of anthocyanin in *Arabidopsis* and *Nicotiana tabacum* [11,12]. In grapevine, *VlMYBA1-2*, *VlMYBA1-3*, and *VlMYBA2*, three R2R3-MYB ranscription factors, isolated from *Vitis labruscana* promoted anthocyanin accumulation by controlling the expression of UFGT in the latest phases of fruit maturation [13]. On the other hand, the transcription factor *VvMYB14* was implied in the regulation of the biosynthesis of the resveratrol phytoalexin in grapevine [14,15]. While two other MYBs, *VvMYB5a* and *VvMYB5b*, may indirectly activate anthocyanin synthesis through activation of upstream biosynthesis genes in grapevine [16,17]. In addition, other R2R3 MYB transcription factors regulating the flavonol pathway have been identified in grapevine, such as *VvMYBPA2*, *VvMYBF1*, and *VvMYBPAR* [18,19,20]. On the other hand, some R2R3 MYB proteins, acting as the repressors, also participate in the regulation of flavonoid biosynthesis [21,22,23,24]. In *Arabidopsis*, *AtMYBL2* and *AtMYB60* inhibit anthocyanin biosynthesis in *Arabidopsis* or *Lactuca sativa* plants [21,22]. The strawberry (Fragaria × ananassa) *FaMYB1* gene repressed anthocyanin synthesis when it was expressed in tobacco [23], while in Chinese narcissus (*Narcissus tazetta*), *NtMYB2* significantly reduced the accumulation of anthocyanins by activating the transcript levels of anthocyanin biosynthesis structural genes [24].

Repressors of the flavonoid pathway play a key role in moderating the extent and distribution of anthocyanin–derived pigments in plant tissues. However, knowledge about the negative regulation of proanthocyanin or anthocyanin biosynthesis is sparse. As a *R2R3-MYB* genes with the C2 repressor motif, *VvMYBC2L2* was recently studied with respect to its temporal and spatial expression profile in grapevine, but its function in anthocyanin synthesis was not tested [25]. In this paper, we present the transcriptional and functional characterization of the *VvMYBC2L2* gene. The bioinformatics analysis, subcellular localization expression pattern, and ectopic expression in tobacco suggest that *VvMYBC2L2* functions as a transcriptional repressor of anthocyanin biosynthesis in grapevine.

## 2. Results

### 2.1. VvMYBC2L2 Sequence Analysis

The 681bp coding sequence of *VvMYBC2L2* was cloned from cDNA isolated from grapevine ‘Yatomi Rose’ flowers. It encodes a putative protein of 227 amino acids with a predicted protein molecular weight of 25.1 kDa (GenBank accession No. ACX50288). Chromosome location of the *VvDRL1* sequence in the *V. vinifera* cv. Pinot Noir clone P40024 genome indicated that *VvMYBC2L2* is mapped on chromosome 17, spanning 1057 bp, including 2 intron (84 bp and 292 bp) and 3 exon (179, 131, and 371 bp) (Figure 1a). Alignment of the *VvMYBC2L2* protein with other MYB proteins showed that the *VvMYBC2L2* contained a N-terminal R2R3-DNA binding domain of 108 amino acids (Position 21–128) with a motif, [D/E]Lx2[R/K]x3Lx6Lx3R, for interaction with a basic helix-loop-helix (bHLH) protein [26] (Figure 1b). Like AtMYB4, *VvMYBC2L2* also contained a C-terminal C2 motif (LN[D/E]L[G/S]), which is suggested to be involved in transcriptional repression [5,21]. In addition, a new motif TLLLFR at C-terminal was identified, which was part of the EAR repressor motif (Figure 1b). Due to the divergence in the C-terminal domains, members of R2R3 MYB proteins belong to different branches [27]. Phylogenetic tree indicated that *VvMYBC2L2* belongs to subgroup 4 of R2R3 MYB family (Figure 1c). *VvMYBC2L2* was closely related to *VvMYBC2L1*, *VvMYBC2L3* and MdMYB111. Moreover, compared with other proteins, *VvMYBC2L2* showed high homology with *VvMYBC2L1* (57.4% similarity), *VvMYBC2L3* (54.3% similarity) in grapevine, MdMYB111 (54.1% similarity) in apple and FaMYB1 (48.9% similarity) in strawberry. There results suggested that *VvMYBC2L2* encodes a putative MYB transcriptional repressor in grapevine.

### 2.2. VvMYBC2L2 Is Localized to the Nucleus

Amino-acid sequence analysis indicated that *VvMYBC2L2* contains a nuclear localization signal (NLS), KPDVKRGNFGEDEEDLIIKLHALLGNRWSLI (position 73–104) (http://nls-mapper.iab.keio.ac.jp/cgi-bin/NLS_Mapper_form.cgi). To confirm the subcellular localization of *VvMYBC2L2*, we created a construct harboring a *VvMYBC2L2-GFP* fusion gene. Figure 2 shows that the *VvMYBC2L2* fused protein was exclusively localized in the nucleus, whereas the control GFP protein was distributed throughout the cytoplasm and cell wall. The results indicated that *VvMYBC2L2* is localized to the nucleus.

### 2.3. Transcript Profiles of VvMYBC2L2

The expression pattern of *VvMYBC2L2* in different tissues and organs was carried out in grapevine by RT-qPCR (Figure 3). In grapevine, the expression level of *VvMYBC2L2* was highest in the root, similar to the expression levels of *VvDRF1*, *VvPAL1* and *VvLDOX1*, but higher than that of *VvLAR1* and *VvUFGT1*. In addition, *VvMYBC2L2* was expressed at a higher level in the flower than in stems, leaves and tendrils, patterns similar to those of *VvDFR1*. A difference was that there was a greater transcript abundance in tendrils in *VvLAR1*, *VvDOX1* and *VvUFGT1* compared with *VvMYBC2L2*. In the developing fruit, the transcript abundance of *VvMYBC2L2* in skins was highest at 2 weeks after flowering (WAF) and decreased thereafter, similar to the temporal variation in transcript abundance of *VvPAL1* and *VvLAR1*. In contrast to these transcript profiles, the transcript abundances of *VvDFR1*, *VvDOX1* and *VvUFGT1* in grape skins were lowest at 2 WAF and increased thereafter, reaching the highest level in 8 WAF.

We obtained the 1238-bp *VvMYBC2L2* promoter from Yatomi Rose leaves. The *cis*-regulatory elements were predicted using the PlantCARE database (http://bioinformatics.psb.ugent.be/webtools/plantcare/html/). In addition to some typical promoter elements, such as the TATA and CAAT boxes, many *cis*-acting elements associated with anthocyanin biosynthesis, such as the G-box, P-box, chs-CMA2a and chs-Unit 1 mL, were found in the *VvMYBC2L2* promoter. The organ-based expression patterns under the control of *VvMYBC2L2* promoter were monitored using GUS reporter gene activity in tobacco. In tobacco seedlings, GUS activities were high in roots and stems, but low in leaves (Figure 3ci1,ci2). In flowers, a high level of GUS expression was found in the calyx at the early stage of flowering, with even greater activity in anthers and stigmas (Figure 3ci3). In fruit, there was a very low level of GUS expression in the seed coat (Figure 3ci4), but high levels in the seed (Figure 3ci5).

### 2.4. Overexpression of VvMYBC2L2 in Tobacco Represses the Pigmentation of Petals

To investigate whether the *VvMYBC2L2* gene is involved in anthocyanin biosynthesis, we produced transgenic tobacco plants overexpressing *VvMYBC2L2*. Three T_1_ lines were chosen by qPCR analysis on the basis of different accumulation levels of *VvMYBC2L2* transcripts (Figure 4a). There was no difference in apparent morphology, such as the size of leaves, plant height and fertilization capacity, between the wild-type and transgenic tobacco plants. However, there was a significant difference in flower color. Compared with the dark pink of the wild type plant, there was a prominent loss of petal pigmentation of petal in the *VvMYBC2L2*-overexpressing plants, ranging from white petals with pink dots or sectors to completely white petals (Figure 4b). To confirm that loss of color in the flower was the result of decreased synthesis of pigments derived from the flavonoid pathway, the total anthocyanin concentration in the flowers was determined. The total anthocyanin accumulation in transgenic tobacco petals was significantly decreased, showing a 94–99% decrease in anthocyanin concentration compared with that in the wild type (Figure 4c). These results suggest that *VvMYBC2L2* negatively regulates anthocyanin biosynthesis.

### 2.5. VvMYBC2L2-Regulated Flavonoid Biosynthetic Gene Expression in Tobacco Flowers

To further elucidate the mechanism underlying the loss of pigmentation in transgenic *VvMYBC2L2*-overexpressed tobacco flowers, we measured the expression level of genes involved in the flavonoid biosynthetic pathway by using qPCR (Figure 5). The genes selected were chalcone isomerase (*NtCHI*), *NtCHS* and flavonol synthase (*NtFLS*) from flavonol biosynthesis, and *NtDFR*, *NtLAR*, anthocyanidin reductase (*NtNAR1* and *NtANR2*), anthocyanidin synthase (*NtNAS2*) and *NtUFGT* from anthocyanin biosynthesis. Of these nine structural genes, the expressions of seven, *NtCHI*, *NtCHS*, *NtDFR*, *NtLAR*, *NtANR2*, *NtANS2* and *NtUFGT*, were down-regulated in transgenic tobacco petals compared with the wild type. In particular, the abundance of *NtCHS*, *NtDFR*, *NtLAR* and *NtUFGT* transcripts in transgenic tobacco showed remarkable reductions, namely 96.2–96.4%, 93.9–94.8%, 94.1–94.8% and 96.4–97.1% decreases, respectively, compared with the wild type. The transcript abundances of NtANR1 and NtNAS2 were weakly suppressed in the transgenic plants, with 11.8–26.3% and 5.6–20% reductions, respectively, compared with the wild type. On the other hand, overexpressing *VvMYBC2L2* in tobacco plants enhanced the expression of *NtFLS*. In addition, as the regulatory genes, the transcript abundances of *NtAN1a* and *NtAN1b* were almost completely suppressed in the transgenic plants, with 98.8–98.9% and 98.6–98.8% reductions, respectively, compared with the wild type.

Taken together, these results indicated that *VvMYBC2L2* functions as a repressor of anthocyanin biosynthesis.

## 3. Discussion

In grapevine, 108 R2R3-MYB transcription factors have been identified from the genome of the PN40024 genotype of *Vitis vinifera* cv. Pinot Noir, which play an important role in the flavonoid pathway [29]. In this work, we reported the isolation and characterization of a R2R3-MYB transcription factor, named *VvMYBC2L2*. *VvMYBC2L2* belongs to subgroup 4 of the R2R3-MYB family (Figure 1b). The members of this group, such as *AtMYB4*, *MdMYB111* and *VvMYBC2L3*, all contain the C2 motif, which functions in transcriptional repression [30]. There was also a putative TLLLFR-type repressor motif at the C-terminnus of *VvMYBC2L2*, which was also found in the flavonoid repressor *AtMYBL2* [21]. Compared with other R2R3 MYB proteins, *VvMYBC2L2* has strong similarities with *VvMYBC2L3*, *MdMYB111* and *FaMYB1* which negatively regulate the flavonoid pathway [23,25,31]. These results suggested that *VvMYBC2L2* may play a role as an active transcriptional repressor of flavonoid biosynthesis in grapevine.

In grape vine ’Corvina’, *MYBC2L2* showed very low expression levels in almost all organs including the berry and seed [25]. However, *VvMYBC2L2* showed unique spatiotemporal specificity among anthocyanin regulators in grapevine ’Yatomi Rose’. Abundant transcripts were detected in roots and flowers, with particularly high expression in berry skins, reaching maximum levels at the pre-veraison stages (2 WAF) and decreasing thereafter during berry development. The same expression profile was shown by *VvLAR1* and *VvPAL1*, implying that *VvMYBC2L2* was involved in the flavonoid pathway. Most of the other structural genes of the flavonoid biosynthesis pathway, such as *VvDFR1* and *VvDLOX1,* showed a different trendwith the transcript abundance being higher at flowering and during early berry development, then declining after flowering, and increasing again at véraison (Figure 3). In particular, *VvUFGT1* showed an expression pattern opposite to that of *VvMYBC2L2*, highly induced in the skin after veraison. UFGT is a key enzyme, catalyzing the final step of the anthocyanin biosynthesis pathway [32]. These findings suggested that *VvMYBC2L2* may have a role as a negative regulator of anthocyanin synthesis in grapevine.

By means of heterologous expression in tobacco, we confirmed the regulatory function of *VvMYBC2L2* in anthocyanin synthesis. Constitutive expression of *VvMYBC2L2* repressed the accumulation of anthocyanins in transgenic tobacco flowers, but did not change the phenotype during vegetative development. *VvMYBC2L1* and *VvMYB4*-like transgenic tobacco plants also showed the loss of pigments in the flower petals [33]. Similar results were obtained in genes from other species, including *FaMYB1* [23], *AtMYB60* [34], *GtMYBIR1*, and *GtMYBIR2* [35]. These results support the role of *VvMYBC2L2* as a flavonoid repressor.

Expression analyses also confirmed that overexpression of *VvMTBC2L2* causes drastic down-regulation of flavonoid-related genes, with *NtCHS*, *NtDFR*, *NtLAR* and *NtUFGT* transcripts all being notably suppressed. CHS is the first key enzyme in flavonoid biosynthesis, catalyzing condensation of p-coumaroyl-coenzyme A and malonyl-CoA to form chalcone. In dahlias, suppression of CHS resulted in white flowers [36], while, in *Petunia hybrid*, constitutive expression of an ‘anti-sense’ CHS gene inhibits flower pigmentation [37]. DFR participates in anthocyanin biosynthesis by catalyzing the conversion of dihydroflavonol to leucoanthocyanidins. Constitutive expression of the *Vitis bellula* gene *VbDFR* increased anthocyanin accumulation in tobacco [38], while, over-expression of the grapevine transcription factor genes carrying the C2-motif repressor, *VvMYBC2L1* and *VvMYBC2L3*, significantly decreased the transcript level of DFR in *Petunia hybrida* [25]. UFGT is involved in glycosylation of anthocyanidins to form anthocyanins, and was specific induced in red cultivars of *V*. *vinifera* but not white cultivars. In Chinese narcissus, *NtMYB2* negatively regulated anthocyanin biosynthesis by down-regulating expression of the flavonoid genes, particularly UFGT [24]. Analysis indicated that suppression of specific flavonoid biosynthetic genes could repress the accumulation of anthocyanins. The decrease in floral pigmentation when *VvMYBC2L2* was expressed in tobacco might be achieved by regulating these flavonoid biosynthetic genes. Moreover, the transcript levels of the regulatory genes, *NtAN1a* and *NtAN1b*-in *VvMYBC2L2*-overexpressed transgenic tobacco plants were also sharply reduced. *NtAN1a* and *NtAN1b*, two members of the basic helix-loop-helix (bHLH) transcription factor family, positively regulate the anthocyanin pathway in tobacco flowers [39]. It is reported that combinatorial interactions between MYB and bHLH transcription factors within the MBW (MYB-bHLH-WD40) complex are crucial for the regulation of the flavonoid pathway. *VvMYBC2L2* possessed the bHLH-interacting signature in the R3 repeat (Figure 1b), so we speculated that *VvMYBC2L2* might negatively regulate the flavonoid pathway through the participation of the MBW complex.

In summary, we confirmed that the *VvMYBC2L2* gene plays a role as a negative regulator in anthocyanin biosynthesis in grapevine. To date, negative regulation of anthocyanin biosynthesis in plants is largely unknown, our findings will provide a new insight into such regulatory mechanisms.

## 4. Materials and Methods

### 4.1. Plant Material and Growth Conditions

*Vitis vinifera* cv. Yatomi Rose (Red table grape cultivar) was grown in the grape germplasm resource orchard of Shandong Institute of Pomology, Taian, Shandong, China. Samples collected from roots, stems and leaves were taken from grapevine plants (*Vitis vinifera* cv. Yatomi Rose) and samples collected from flowers were collected 2 days before flowering (Pre-capfall pollinization period). Samples taken from fruits corresponded to the fruits skin removed from fruits at different developmental stages: 2, 3, 5, 7 and 8 weeks after flowering (WAF). Three biological replicates were collected from three different plants. All collected samples were frozen in liquid nitrogen and stored at −80 °C. Tobacco plants (*Nicotiana tabacum* cv. NC89 and *Nicotian benthamiana*) were grown in vitro on MS medium under a 16/18 h photoperiod at 25 °C.

### 4.2. Isolation of the VvMYBC2L2 Promoter Sequence

DNA from the grapevine leaves was extracted using the CTAB method. Based on the grapevine genome sequence, we designed the primers with the forward primer 5′-TACCACCGGAAAAGTACAATACCATCT-3′ and reverse primer 5′-GGCGATAGAGAAAGACCGTAGAG AG-3′ (Appendix A). Promoter amplification was performed using the PrimerSTAR GXL DNA Polymerase (Takara, Dalian, China) with the PCR reactions: 2min at 94 °C, 30 cycles of 30 s at 94 °C, 30 s at 56 °C, 1.5 min at 72 °C, then 10 min at 72 °C. The PCR fragment was cloned into pMD18T vector (Takara) and the colonies were sequenced by Sangon Biotechnology Company (Shanghai, China).

### 4.3. RNA Isolation and Expression Analysis

Total RNA from the grapevine was extracted using the method described by Reid et al. [40]. The integrity of extracted RNA was checked on 1% agarose gels to ensure that the 28S and 18S were clear without tailing and that the 28S:18S ratio was 2:1. The RNA concentration was determined using an Ultramicro Spectrophotometer (P-330-31, Implen, Munich, Germany). The cDNA first strand was synthesized from 1 µg total RNA using the PrimerScript^TM^ II 1st Strand cDNA Synthesis Kit (Takara). Quantitative RT-PCR was run on IQ5 real time PCR cycler (Bio-Rad Laboratories, Hercules, CA, USA) with SYBR^®^ Green I dye. Reactions were the following thermal profile: 3 min at 94 °C, 40 cycles of 5 s at 94 °C, 30 s at 58 °C. The relative mRNA rations were calculated as 2^−ΔΔCT^ [41]. The transcript levels of target genes were normalized against *VvActin* for the grape samples and *NtActin* for *Nicotiana* samples [40,42]. The data are presented as the mean value of three biological replicates.

### 4.4. Subcellular Location of the VvMYBC2L2

The coding region of *VvMYBC2L2* without the termination codon was amplified using the PrimerSTAR GXL DNA Polymerase (Takara, Shiga, Japan) with the forward primer 5′-GGTCTAGAATGG TTGCAATGAGGAAGCCTGCAG-3′ and reverse primer 5′-GGGGTACCTCCAAAAAGAAGTAGA GTTGTAAAG-3′. Reaction was the following thermal profile: 2 min at 94 °C, 30 cycles of 30 s at 94 °C, 30 s at 56 °C, 1 min at 72 °C, then 10 min at 72 °C. The resulting fragments were digested with XbaI and KpnI and then cloned into the vector pBI221-GFP to generate *pBI221-VvMYBC2L2-GFP* constructs. The polyethylene glycol-mediated transfection of *A. thaliana* mesophyll protoplasts was conducted according to previously described protocols [43]. The location of the fusion protein was observed 16 h after transformation using a confocal microscope (LSM510; Carl Zeiss Thornwood, New York, NY, USA).

### 4.5. Construct Assembly and Plant Transformation 

The coding sequence of *VvMYBC2L2* was amplified by PCR. The resulting fragment were digested by Sall and Sacl and sub-cloned into vector pRI101-AN. The 1229-bp promoter of *VvMYBC2L2* was inserted into the vector pCAMIA1391z to generate the construct *VvMYBC2L2_pro_:GUS*. These constructs were transferred into *Agrobacterium tumefaciens* strain GV3101 by electrofusion. The transformation of the leaf discs and the regeneration of the transgenic tobacco plants were carried out according to the protocol described by Horsch et al. [44]. Transgenic plants were selected on MS medium with kanamycin (100 mg·L^−1^) or hygromycin (25 mg·L^−1^) added. T_1_ generation tobacco plants were used for further analysis.

### 4.6. Determination of Total Anthocyanin Content Concentration

The total anthocyanin concentration in tobacco flowers was determined as previously described [45]. The fresh flower were ground in liquid nitrogen and extracted with methanol (containing 1% HCl). The absorbance of supernatants was determined at 530 nm and 657 nm using a UV-Visible Spectrophotometer (UV2600, Shimadzu, Kyoto, Japan). The concentration of anthocyanin was determined using the following equation: (A_530_ − 0.25 × A_657_) × FW^−1^. All samples were measured in triplicate and as three independent biological replicates.

### 4.7. Histochemical GUS Analysis

Histochemical GUS analysis was performed according to the method of Jefferson et al. with minor modifications [46]. The samples were put into acetone (90%) for 20–30 min at 4 °C, and rinsed in wash buffer for 30 min at 4 °C, and then immersed in GUS staining solution and a vacuum applied until the samples sunk. After incubation at 37 °C overnight, the samples were washed with 70% ethanol until the chlorophyll had faded completely. The GUS staining solution (100 mM NaH_2_PO_4_, 100 mM Na_2_HPO_4_, 10 mM EDTA, 0.5 mM K_3_Fe(CN)_3_, 0.5 mM K_4_Fe(CN)_4_, Triton-X 0.1%, X-GLuc 0.5mg·L^−1^, pH 7.0) was stored at −20 °C.

## Figures and Tables

**Figure 1 molecules-24-00092-f001:**
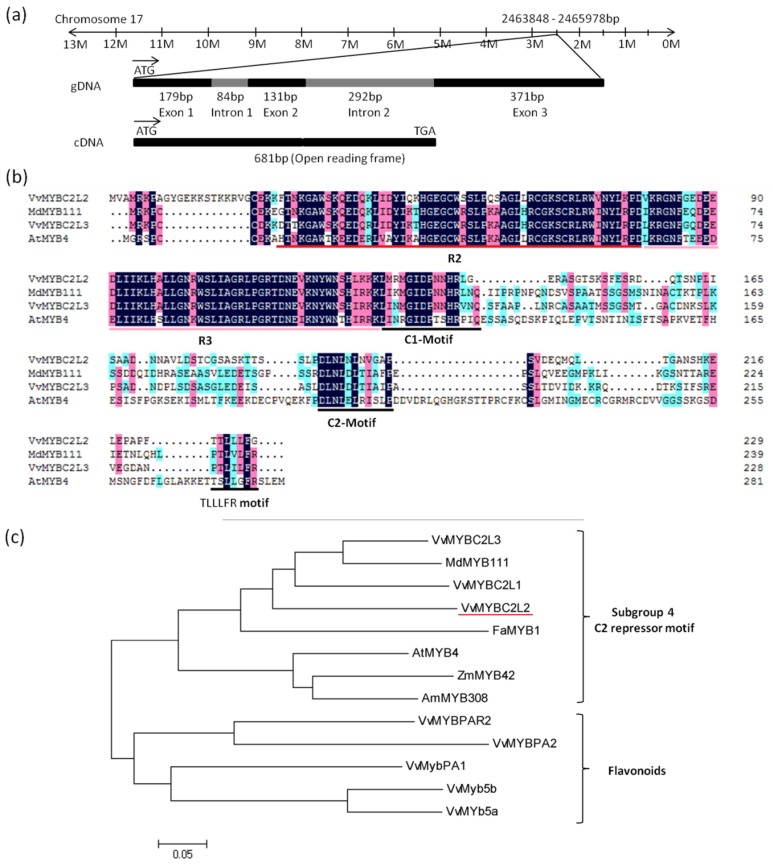
Features of the predicted *VvMYBC2L2* protein. (**a**) The genomic sequence of *VvMYBC2L2*. The 1057 bp gDNA of *VvMYBC2L2* contains three exons and two introns. (**b**), Alignment of the deduced amino acid sequence of *VvMYBC2L2* with other repressor R2R3-MYB proteins. R2 and R3 are the two repeats of the MYB DNA-binding domain. C1 motif, C2 motif and TLLLFR motif are also boxed at the C-terminus position. (**c**) Phylogenetic analysis of R2R3-MYB transcription factors. The phylogenetic tree was generated using the neighbor-joining method by the MEGA 5.0 software (Tokyo Metropolitan University, Hachioji, Tokyo, Japan) [28]. GenBank accession numbers are as follows: *AtMYB4* (AF062860), *AmMYB308* (P81393), *FaMYB1* (AF401220), *MdMYB111* (HM122615)*, VvMYBC2L1* (ABW34393), *VvMYBC2L2* (ACX50288), *VvMYBC2L3* (AIP98385), *VvMYB5a* (AAS68190), *VvMYB5b* (AAX51291), *VvMYBPA1* (CAJ90831), *VvMYBPA2* (ACK56131), *VvMYBPAR2* (BAP39802).

**Figure 2 molecules-24-00092-f002:**
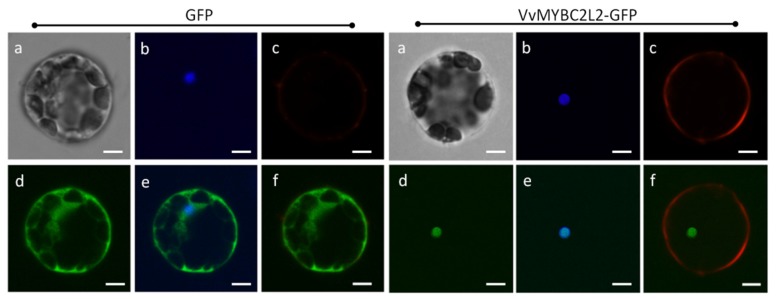
Subcellular localization of the *VvMYBC2L2* protein. Transient expression of the VvMYBC2L2-GFP fusion protein was performed in *A. thaliana* mesophyll protoplasts. The fluorescent signal was detected by confocal laser-scanning microscopy. Bright field (**a**), DAPI (**b**), Dil (Lipophilic membrance dye) (**c**), GFP (**d**), GFP + DAPI merge (**e**), GFP + Dil merge (**f**). Bars correspond to 10 μm.

**Figure 3 molecules-24-00092-f003:**
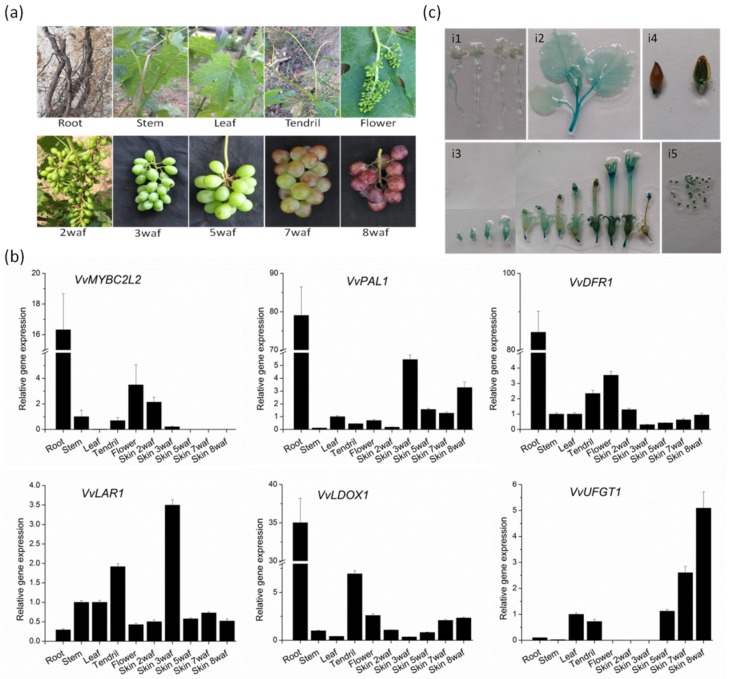
Transcript abundance of grapevine genes associated with proanthocyanidin biosynthesis. (**a**) The different organs (roots, stems, leaves, tendrils and flowers) and berries during berry development of grapevine. Data points during skin development are showed as weeks after flowering (WAF). (**b**) The expression level of flavonoid-related biosynthetic genes. Transcript levels were measured by quantitative real-time PCR analysis and the expression data were normalized against the expression of *VvActin*, with the bar representing the mean of the three biological and three technical replicates SD. (**c**) GUS staining patterns for the *VvMYBC2L2* promoter in tobacco plants. i1, 7 days old seedling; i2, leaves and stem at 8 weeks old; i3, flowers at the early, middle and late development period; i4, fruit; i5, seed.

**Figure 4 molecules-24-00092-f004:**
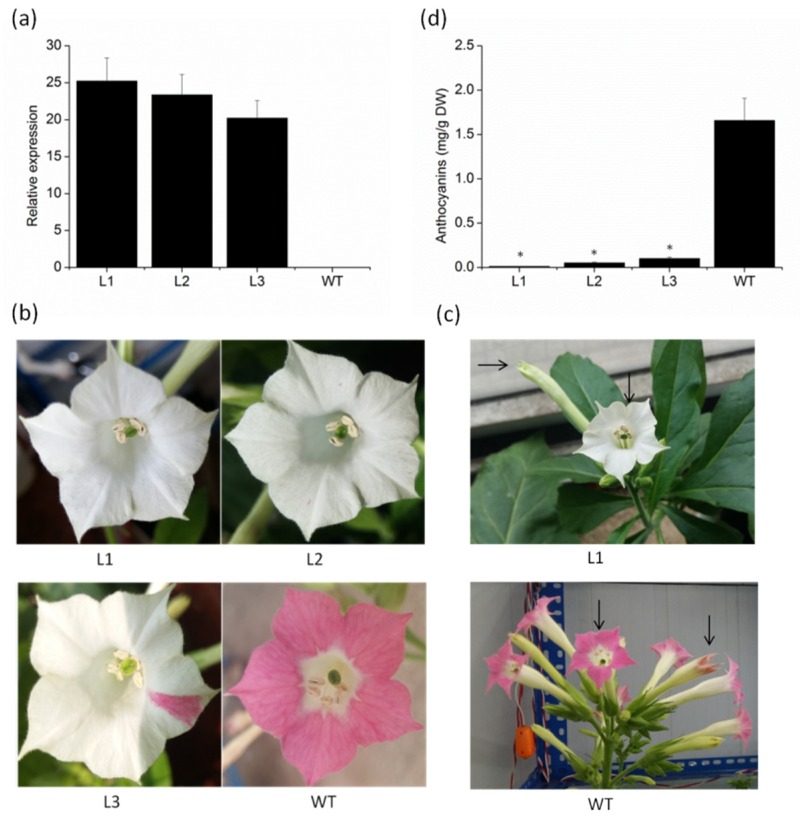
The phenotypes and anthocyanin concentrations of *VvMYBC2L2*-overexpressed transgenic tobacco flowers. (**a**) Expression levels of *VvMYBC2L2* transcripts in wild-type (WT) and transgenic plants. L1, L2, L3 represent three transgenic tobacco lines. Bars represent mean ± SD. Transcript levels were measured by RT-PCR analysis and the expression data were normalized against the expression of *NtActin*. (**b**) Typical floral phenotypes of WT and transgenic plants. (**c**) The color of transgenic and WT tobacco flowers at different flower stages. (**d**) The anthocyanin concentrations in WT and transgenic petals. Anthocyanins were extracted with methanol containing 1% (*v*/*v*) hydrochloric acid, and the absorbance of the solution was measured with a UV-Visible Spectrophotometer. Bars represent mean ± SD. Significant differences from the WT were confirmed by ANOVA and Tukey’s test (* *p* < 0.05).

**Figure 5 molecules-24-00092-f005:**
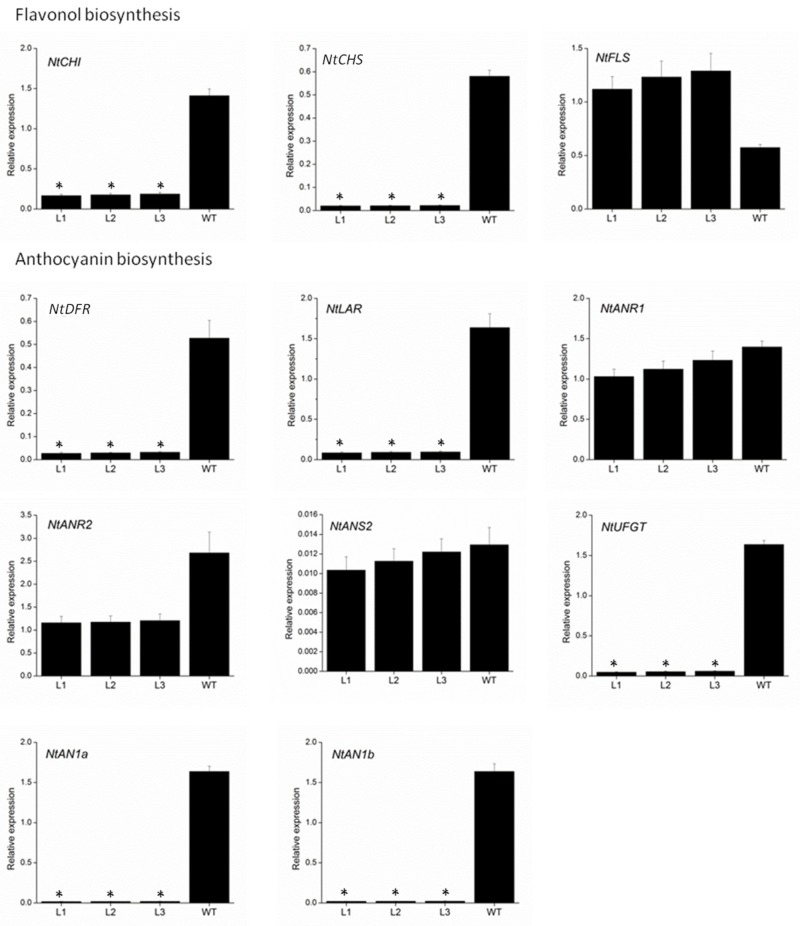
Quantitative analyses of transcript levels of flavonoid-related biosynthetic and regulatory genes in petals of wild-type and transgenic tobacco. The expression data were normalized against the expression of *NtActin*. Bars represent mean ± SD. Significant difference from the WT was confirmed by ANOVA and Tukey’s test (* *p* < 0.05).

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
