# Peer review of "A R2R3-MYB Transcription Factor, *VvMYBC2L2*, Functions as a Transcriptional Repressor of Anthocyanin Biosynthesis in Grapevine (*Vitis vinifera* L.)"

_molecules, 2018, doi:10.3390/molecules24010092_

Reviewer 1 Report

molecules-401697-peer-review-v1

Full Title: A R2R3-MYB transcription factor, VvMYBC2L2, functions as a transcriptional repressor of anthocyanin biosynthesis in grapevine (Vitis vinifera)

General comments to the manuscript:

The present manuscript describes a study that deals with the identification of a new R2R3-MYB transcription factor member in grapevine. Besides the characterization at sequence level the authors here present results achieved by applying different approaches that allowed them to confirm the in silico results regarding cellular location, and to propose the involvement of that transcription factor on regulation of anthocyanin biosynthesis genes (achieved by gene expression and functional analysis).

The manuscript is very interesting and would merit of publication in the Molecules journal; however, there are some points that need to be improved before final acceptance, which I think that will contribute to increase the quality of manuscript and consequently the number of citations.

I would recommend to improve the manuscript making the changes suggested below. Additionally, a revision of the language must be considered by a native English speaker due to the existence of several typographical and grammatical errors.

I agree with major revision.

 Abstract

The abstract must be improved considering that the use of “was involved” in line 17 is too much affirmative and the authors must write the sentence considering that the results achieved are giving only an indication of the involvement.

In line 17: “VvMYBC2L2 was a nuclear protein, and strongly expressed in root, flower and seed, but decreased with the fruit development in grapevine”. This results must be introduced describing the approach followed and written in the simple present form. Additionally, when authors write “strongly expressed” after describing the enzyme location, it lead reader to think that the expression is related with the enzyme and not with the gene. This must be rewritten.

 Introduction

The way that authors structure the sentences related with functional analysis is not correct and must be changed. There are several examples, and some of them I am presenting below:

Line 38:” In Arabidopsis, PRODUCTION ANTHOCYANIN PIGMENT1 (PAP1) and PAP2 induced the

 accumulation of anthocyanin in Arabidopsis and Nicotian tabacum [8, 9].”

Line 47:” Arabidopsis, AtMYBL2 and AtMYB60 inhibit anthocyanin biosynthesis in the Arabidopsis or lettuce plant [16, 17].”

Line 48: “In strawberry, FaMYB1 could repress anthocyanin synthesis when was expressed in  tobacco [18]”

Line 200: “In grapevine, VvMYBC2L1 and VvMYB4-like transgenic tobacco plants also showed the loss of the pigment in the flower petal.”

To rephrase those sentences the authors can follow the example they wrote in line 210 “Constitutive expression of the V. bellila VbDFR increases anthocyanin accumulation in tobacco [31].”

Please remove the expression “so on” along the text.

Complete the information given at sentence in line 46 “On the other side, some R2R3 MYB proteins, acting as repressors, participate in the regulation of flavonoid biosynthesis by …”

Line 55: “…grapevine, but its function in anthocyanin synthesis was not tested.” Include  related references.

 Results

Line 67: remove italic from VvMYBC2L2 since the information is related with the putative protein.

Line 69: Don’t start a sentence with “And”. Please revised that along all the text, there are other places where the sentence starts in that way.

Line 70: Rewrite that sentence. It is not clear why the AtMYB4 appears at the end of the sentence.

Line 70: Rewrite sentence to “Moreover, a new motif at the C-terminal was identified, the TLLLFR, …”

Line 72: “Phylogenetic tree indicated that VvMYBC2L2 belongs to the 4 subgroup of R2R3 MYB family”. The authors must include a short explanation about this analysis, the sequences used and the aim of that analysis. In my point of view the authors only can say that the isolated sequence belong to subgroup 4 if sequences from all the groups are included in the alignment to further construct the dendrogram. As far as I can understand only sequences from the subgroup 4 were included. In line 212 of discussion section the authors present some results previously published by other authors, in which are included the VVMYBC2L1 and VVMYBC2L3. This last sequence was considered for dendrogram construction but the  VVMYBC2L1 was not, why? I would suggest to complete this analysis including sequences from all subgroups.

Line 73: Rephrase that sentence.

The legend of Figure 1 must be completed including the information related with the nature of sequences used (nucleotide or peptide), if it was considered the complete or partial sequence, and how many replicates were considered (1000 ?).

In line 89 I suggest to change “To determine” by “To confirm”, because the in silico analysis have suggested a cellular location and the wet experiment will allow authors to get the confirmation.

Line 97 in legend of Figure 2: what means Dil?

Line 101: The reference to expression analysis by qPCR must be changed to RT-qPCR (reverse transcription quantitative real-time polymerase chain reaction) (see MIQE paper, Bustin et al., 2009). All the technical aspects of qPCR experiments should fit this paper.

Rephrase sentence from line 100-102.

The authors must complete the information given in line 103-104 “A difference…” comparing with what?

In line 106 include the complete name of WAF. I think this is the first time that this acronym appears but the authors must confirm that.

Line 108: correct the word “creased”, I think that authors want to say “increased thereafter and reached the highest level at 8 WAF”.

Line 120, in legend of Figure 3 the authors must include the complete name of PA. The authors must consider that the first time they use an acronym they must include the complete name, this must also be applied for the genes.

In figure 5 the graph of named VvCHS must be changed to NtCHS. Additionally I would recommend to revise the results of statistical analysis for the NtFLS gene since no significant differences were included.

Discussion

The sentence in line 182 must be rephrased, like “These results suggested that VvMYBC2L2 may play an active role in transcriptional regulation of gene expression acting as repressor.”

 Materials and Methods

 Change sentences 232-2333 and complete the information required “Different organs including roots, stems, leaves and flowers were sampled. Samples collected from roots, stems and leaves were taken from ….., and samples collected from flowers were collected …. days after flowering (it was the complete flower??). Samples taken from fruits corresponded to the fruits skin removed from fruits at different developmental stages: 2, 3, 5, 7 and 8 weeks after flowering (WAF). Three biological replicates were collected from three different plants (right?).”

Remove sentence “The skin samples were separated from the rest of the berry”

The title “Isolation of genomic DNA and promoter of the VvMYBC2L2” must be changed to “Isolation of the VvMYBC2L2 promoter sequence”

This point is highly incomplete, there are missing the primers used, the PCR conditions, the procedure for cloning, sequencing, and sequence analysis.

Change sentence in line 243 “Total RNA was extracted from each sample using the method…”

The authors are here reporting the use of a single gene as reference gene. Nowadays the use of a single gene as reference gene is not easily acceptable, so the authors must support the use of a single gene with some previous results or with a reference that show the stability of this gene in a similar experiment with grapevine.

In the section of RT-qPCR there are missing the information related with RNA analysis (integrity and quality), cDNA synthesis, technical replicates used in RT-qPCR, analysis of primers specificity and efficiency. All this information must be added.

Include in line 251 and 258 the information related with PCR conditions (primers and PCR conditions).

 Author Response

Dear editor and reviewers:

Thanks for your advice. According to the reviewer’ comments, we revised the manuscript and explain point-by-point the details of the revisions. Meanwhile, our manuscript has been edited by the International Science Editing ( http://www.internationalscienceediting.com ).

If you have any questions, please contact us. We look forward to hearing from you soon.

The following is our responses to the reviewer’s comments.

 Point 1: The abstract must be improved considering that the use of “was involved” in line 17 is too much affirmative and the authors must write the sentence considering that the results achieved are giving only an indication of the involvement.

Response 1: We have revised. The revised is “Here, a R2R3-MYB transcription factor, VvMYBC2L2, isolated from Vitis vinifera cultivar Yatomi Rose, may be involved in anthocyanin biosynthesis as a transcriptional repressor.” See line 18.

Point 2: In line 17: “VvMYBC2L2 was a nuclear protein, and strongly expressed in root, flower and seed, but decreased with the fruit development in grapevine”. This results must be introduced describing the approach followed and written in the simple present form. Additionally, when authors write “strongly expressed” after describing the enzyme location, it lead reader to think that the expression is related with the enzyme and not with the gene. This must be rewritten.

Response 2: We have revised. The revised is “VvMYBC2L2 was shown to be a nuclear protein. The gene was shown to be strongly expressed in root, flower and seed tissue, but weakly expressed during the fruit development in grapevine.” See line 19

Point 3: The way that authors structure the sentences related with functional analysis is not correct and must be changed. There are several examples, and some of them I am presenting below:

Line 38:” In Arabidopsis, PRODUCTION ANTHOCYANIN PIGMENT1 (PAP1) and PAP2 induced the

 accumulation of anthocyanin in Arabidopsis and Nicotian tabacum [8, 9].”

Line 47:” Arabidopsis, AtMYBL2 and AtMYB60 inhibit anthocyanin biosynthesis in the Arabidopsis or lettuce plant [16, 17].”

Line 48: “In strawberry, FaMYB1 could repress anthocyanin synthesis when was expressed in  tobacco [18]”

Line 200: “In grapevine, VvMYBC2L1 and VvMYB4-like transgenic tobacco plants also showed the loss of the pigment in the flower petal.”

To rephrase those sentences the authors can follow the example they wrote in line 210 “Constitutive expression of the V. bellila VbDFR increases anthocyanin accumulation in tobacco [31].”

Response 3: We have revised. The revised is following:

In Arabidopsis, AtMYBL2 and AtMYB60 inhibit anthocyanin biosynthesis in Arabidopsis or Lactuca sativa plants [18, 19]. The strawberry (Fragaria × ananassa) FaMYB1 gene repressed anthocyanin synthesis when it was expressed in tobacco [20], while in Chinese narcissus (Narcissus tazetta), NtMYB2 significantly reduced the accumulation of anthocyanins by activating the transcript levels of anthocyanin biosynthesis structural genes [21].” See line 56.

VvMYBC2L1 and VvMYB4-like transgenic tobacco plants also showed the loss of pigments in the flower petals [29]. Similar results were obtained in genes from other species, including FaMYB1 [20], AtMYB60 [30], GtMYBIR1, and GtMYBIR2 [31].” See line 237

Point 4: Please remove the expression “so on” along the text.

Response 4: we have removed the expression “so on”.

Point 5: Complete the information given at sentence in line 46 “On the other side, some R2R3 MYB proteins, acting as repressors, participate in the regulation of flavonoid biosynthesis by …”

Response 5: We have revised. The revised is “On the other hand, some R2R3 MYB proteins, acting as the repressors, also participate in the regulation of flavonoid biosynthesis [18-21]. See line 54.

References:

18.    Matsui, K.; Umemura, Y.; Ohme-Takagi, M., AtMYBL2, a protein with a single MYB domain, acts as a negative regulator of anthocyanin biosynthesis in Arabidopsis. The Plant journal : for cell and molecular biology 2008, 55, (6), 954-67.

19.    Galbiati, M.; Matus, J. T.; Francia, P.; Rusconi, F.; Canon, P.; Medina, C.; Conti, L.; Cominelli, E.; Tonelli, C.; Arce-Johnson, P., The grapevine guard cell-related VvMYB60 transcription factor is involved in the regulation of stomatal activity and is differentially expressed in response to ABA and osmotic stress. Bmc Plant Biol 2011, 11.

20.    Aharoni, A.; De Vos, C. H.; Wein, M.; Sun, Z.; Greco, R.; Kroon, A.; Mol, J. N.; O'Connell, A. P., The strawberry FaMYB1 transcription factor suppresses anthocyanin and flavonol accumulation in transgenic tobacco. Plant J 2001, 28, (3), 319-32.

21.    Anwar, M.; Wang, G.; Wu, J.; Waheed, S.; Allan, A. C.; Zeng, L., Ectopic Overexpression of a novel R2R3-MYB, NtMYB2 from Chinese Narcissus represses anthocyanin biosynthesis in tobacco. Molecules 2018, 23, (4).

Point 6: Line 55: “…grapevine, but its function in anthocyanin synthesis was not tested.” Include related references.

Response 6: The related reference has been added. The revised is “As a R2R3 -MYB genes with the C2 repressor motif, VvMYBC2L2 was recently studied with respect to its temporal and spatial expression profile in grapevine, but its function in anthocyanin synthesis was

not tested [22]. See line 65.

Reference:

22.    Cavallini, E.; Matus, J. T.; Finezzo, L.; Zenoni, S.; Loyola, R.; Guzzo, F.; Schlechter, R.; Ageorges, A.; Arce-Johnson, P.; Tornielli, G. B., The phenylpropanoid pathway is controlled at different branches by a set of R2R3-MYB C2 repressors in grapevine. Plant Physiol 2015, 167, (4), 1448-70.

Point 7: Line 67: remove italic from VvMYBC2L2 since the information is related with the putative protein.

Response 7: we have removed italic.

Point 8: Line 69: Don’t start a sentence with “And”. Please revised that along all the text, there are other places where the sentence starts in that way.

Response 8: we have changed.

Point 9: Line 70: Rewrite that sentence. It is not clear why the AtMYB4 appears at the end of the sentence.

Response 9: AtMYB4 contains a C2 motif which is involved in the transcriptional repression, and VvMYBC2L2 also contains a C-terminal C2 motif. Now, we have rewritten the sentence. The revised is “Like AtMYB4, VvMYBC2L2 also contained a C-terminal C2 motif (LN[D/E]L[G/S]), which is suggested to be involved in transcriptional repression [4, 18].” See line 81.

References:

4.      Dubos, C.; Stracke, R.; Grotewold, E.; Weisshaar, B.; Martin, C.; Lepiniec, L., MYB transcription factors in Arabidopsis. Trends Plant Sci 2010, 15, (10), 573-81.

18.    Matsui, K.; Umemura, Y.; Ohme-Takagi, M., AtMYBL2, a protein with a single MYB domain, acts as a negative regulator of anthocyanin biosynthesis in Arabidopsis. The Plant journal : for cell and molecular biology 2008, 55, (6), 954-67.

Point 10: Line 70: Rewrite sentence to “Moreover, a new motif at the C-terminal was identified, the TLLLFR, …”

Response 10: We have written the sentence. The revised is “In addition, a new motif TLLLFR at C-terminal was identified, which was part of the EAR repressor motif (Figure 1b).” See line 83.

Point 11: Line 72: “Phylogenetic tree indicated that VvMYBC2L2 belongs to the 4 subgroup of R2R3 MYB family”. The authors must include a short explanation about this analysis, the sequences used and the aim of that analysis. In my point of view the authors only can say that the isolated sequence belong to subgroup 4 if sequences from all the groups are included in the alignment to further construct the dendrogram. As far as I can understand only sequences from the subgroup 4 were included. In line 212 of discussion section the authors present some results previously published by other authors, in which are included the VVMYBC2L1 and VVMYBC2L3. This last sequence was considered for dendrogram construction but the VVMYBC2L1 was not, why? I would suggest to complete this analysis including sequences from all subgroups.

Response 11: Due to the divergence in the C-terminal domains, members of R2R3 MYB proteins belonged to different branches [24]. We rebuild the phylogenetic tree, which showed that VvMYBC2L1 belongs to subgroup 4 of R2R3 MYB family. So, we supplement information about the classification of MYB family. The revised is “Due to the divergence in the C-terminal domains, members of R2R3 MYB proteins belonged to different branches [24]. Phylogenetic tree indicated that VvMYBC2L2 belongs to subgroup 4 of R2R3 MYB family (Figure. 1c). VvMYBC2L2 was closely related to VvMYBC2L1, VvMYBC2L3 and MdMYB111.” See line 84

Reference:

24.    Rabinowicz, P. D.; Braun, E. L.; Wolfe, A. D.; Bowen, B.; Grotewold, E., Maize R2R3 Myb genes: Sequence analysis reveals amplification in the higher plants. Genetics 1999, 153, (1), 427-44.

Point 12: Line 73: Rephrase that sentence.

Response 12: we have rephrased the sentence. The revised is “Moreover, compared with other proteins, VvMYBC2L2 showed high homology with VvMYBC2L1 (57.4% similarity), VvMYBC2L3 (54.3% similarity) in grapevine, MdMYB111 (54.1% similarity) in apple and FaMYB1 (48.9% similarity) in strawberry. See line 88.

Point 13: The legend of Figure 1 must be completed including the information related with the nature of sequences used (nucleotide or peptide), if it was considered the complete or partial sequence, and how many replicates were considered (1000 ?).

Response 13: In Figure1, they are features of the predicted VvMYBC2L2 protein. Among them, Figure 1a is the analysis of the genomic sequence of VvMYBC2L2. The revised legend revised in Figure 1 is “Figure 1. Features of the predicted VvMYBC2L2 protein. (a) The genomic sequence of VvMYBC2L2. The 1057 bp gDNA of VvMYBC2L2 contains three exons and two introns. (b), Alignment of the deduced amino acid sequence of VvMYBC2L2 with other repressor R2R3-MYB proteins. R2 and R3 are the two repeats of the MYB DNA-binding domain. C1 motif, C2 motif and TLLLFR motif are also boxed at the C-terminus position. (c) Phylogenetic analysis of R2R3-MYB transcription factors. The phylogenetic tree was generated using the neighbor-joining method by the MEGA 5.0 software.” See line 95.

Point 14: In line 89 I suggest to change “To determine” by “To confirm”, because the in silico analysis have suggested a cellular location and the wet experiment will allow authors to get the confirmation.

Response 14: we have changed. The revised is “To confirm the subcellular localization of the VvMYBC2L2”. See line 108.

Point 15: Line 97 in legend of Figure 2: what means Dil?

Response 15: Dil is lipophilic membrane dye which is localized to cell membrane. We have revised in figure 2. See line 117.

Point 16: Line 101: The reference to expression analysis by qPCR must be changed to RT-qPCR (reverse transcription quantitative real-time polymerase chain reaction) (see MIQE paper, Bustin et al., 2009). All the technical aspects of qPCR experiments should fit this paper.

Response 16: we have changed.

Point 17: Rephrase sentence from line 100-102.

Response 17: we have rephrased the sentence. The revised is “The expression pattern of VvMYBC2L2 in different tissues and organs was carried out in grapevine by RT-qPCR (Figure 3). In grapevine, the expression level of VvMYBC2L2 was highest in the root, similar to the expression levels of VvDRF1, VvPAL1 and VvLDOX1, but higher than that of VvLAR1 and VvUFGT1. See line 120.

Point 18: The authors must complete the information given in line 103-104 “A difference…” comparing with what?

Response 18: The written sentence:” A difference was that there was a grater transcript abundance in tendrils in VvLAR1, VvDOX1 and VvUFGT1 compared with VvMYBC2L2.” See line 126.

Point 19: In line 106 include the complete name of WAF. I think this is the first time that this acronym appears but the authors must confirm that.

Response 19: we have revised. WAF: weeks after flowering. See line 128.

Point 20: Line 108: correct the word “creased”, I think that authors want to say “increased thereafter and reached the highest level at 8 WAF”.

Response 20: We have changed. The revised is “In contrast to these transcript profiles, the transcript abundances of VvDFR1, VvDOX1 and VvUFGT1 in grape skins were lowest at 2 WAF and increased thereafter, reaching the highest level in 8 WAF.” See line 130.

Point 21: Line 120, in legend of Figure 3 the authors must include the complete name of PA. The authors must consider that the first time they use an acronym they must include the complete name, this must also be applied for the genes.

Response 21: We have revised it. PA: proanthocyanidin. See line 145.

Point 22: In figure 5 the graph of named VvCHS must be changed to NtCHS. Additionally I would recommend to revise the results of statistical analysis for the NtFLS gene since no significant differences were included.

Response 22: we are sorry that VvCHS should be the NtCHS. In the figure5, we have revised. See line 198.

Point 23: The sentence in line 182 must be rephrased, like “These results suggested that VvMYBC2L2 may play an active role in transcriptional regulation of gene expression acting as repressor.”

Response 23: We have revised. Taken together, these results indicated that VvMYBC2L2 functions as a repressor of anthocyanin biosynthesis.See line 203.

Point 24: Change sentences 232-2333 and complete the information required “Different organs including roots, stems, leaves and flowers were sampled. Samples collected from roots, stems and leaves were taken from ….., and samples collected from flowers were collected …. days after flowering (it was the complete flower??). Samples taken from fruits corresponded to the fruits skin removed from fruits at different developmental stages: 2, 3, 5, 7 and 8 weeks after flowering (WAF). Three biological replicates were collected from three different plants (right?).”

Response 24: Sorry, we did not describe the sample collection clearly. The revised is “Samples collected from roots, stems and leaves were taken from grapevine plants (Vitis vinifera cv Yatomi Rose) and samples collected from flowers were collected 2 days before flowering (Pre-capfall pollinization period). Samples taken from fruits corresponded to the fruits skin removed from fruits at different developmental stages: 2, 3, 5, 7 and 8 weeks after flowering (WAF). Three biological replicates were collected from three different plants. See line 276.

Point 25: Remove sentence “The skin samples were separated from the rest of the berry”

Response 25: we have removed the sentence.

Point 26: The title “Isolation of genomic DNA and promoter of the VvMYBC2L2” must be changed to “Isolation of the VvMYBC2L2 promoter sequence”

Response 26: we have changed the title. “Isolation of the VvMYBC2L2 promoter sequence” See line 287.

Point 27: This point is highly incomplete, there are missing the primers used, the PCR conditions, the procedure for cloning, sequencing, and sequence analysis.

Response 27: We have supplemented the information. The correct is “Based on the grapevine genome sequence, we designed the primers with the forward primer 5'-TACCACCGGAAAAGTACAATACCATCT-3' and reverse primer 5'-GGCGATAGAGAAAGACCGTAGAGAG-3'. Promoter amplification was performed using the PrimerSTAR GXL DNA Polymerase (Takara, Japan) with the PCR reactions: 2min at 94 oC, 30 cycles of 30 s at 94 oC, 30s at 56 oC, 1.5min at 72 oC, then 10min at 72 oC. The PCR fragment was cloned into pMD18T vector (Takara, Japan) and the colonies were sequenced by Sangon Biotechnology Company (China, Shanghai).” See line 288.

Point 28: Change sentence in line 243 “Total RNA was extracted from each sample using the method…”

Response 28: we have changed the sentence. The correct is “Total RNA from the grapevine was extracted using the method described by Reid et al. [33]” See line 298.

Point 29: The authors are here reporting the use of a single gene as reference gene. Nowadays the use of a single gene as reference gene is not easily acceptable, so the authors must support the use of a single gene with some previous results or with a reference that show the stability of this gene in a similar experiment with grapevine.

Response 29: Gutha (2010) reported that Actin was identified as the stable set of reference genes for normalization of gene expression data obtained from grapevine leaves using the geNorm program. Reidalso (2006) also reported that Actin was the more relevant reference genes in grape berry development studies. So the gene actin was chosen as reference gene. The revised is “The transcript levels of target genes were normalized against VvActin for the grape samples and NtActin for Nicotiana samples [36, 38].” See line 306.

References:

36.    Reid, K. E.; Olsson, N.; Schlosser, J.; Peng, F.; Lund, S. T., An optimized grapevine RNA isolation procedure and statistical determination of reference genes for real-time RT-PCR during berry development. Bmc Plant Biol 2006, 6, 27.

38.    Gutha, L. R.; Casassa, L. F.; Harbertson, J. F.; Naidu, R. A., Modulation of flavonoid biosynthetic pathway genes and anthocyanins due to virus infection in grapevine (Vitis vinifera L.) leaves. Bmc Plant Biol 2010,

Point 30: In the section of RT-qPCR there are missing the information related with RNA analysis (integrity and quality), cDNA synthesis, technical replicates used in RT-qPCR, analysis of primers specificity and efficiency. All this information must be added.

Response 30: Sorry, we did not describe the RNA analysis clearly. The revised is “Total RNA from the grapevine was extracted using the method described by Reid et al. [36]. The integrity of extracted RNA was checked on 1% agarose gels to ensure that the 28S and 18S were clear without tailing and that the 28S : 18S ratio was 2: 1.  The RNA concentration was determined using an Ultramicro Spectrophotometer (Implen P-330-31, Germany). The cDNA first strand was synthesized from 1 µg total RNA using the PrimerScriptTM II 1st Strand c DNA Synthesis Kit (Takara, Japan). Quantitative RT-PCR was run on IQ5 real time PCR cycler (Bio-Rad Laboratories) with SYBR® Green I dye. Reactions were the following thermal profile: 3 min at 94 oC, 40 cycles of 5 s at 94 oC, 30s at 58 oC. The relative mRNA rations were calculated as 2-ΔΔCT [37]. The transcript levels of target genes were normalized against VvActin for the grape samples and NtActin for Nicotiana samples [36, 38]. The data are presented as the mean value of three biological replicates.” See line 298.

Point 31: Include in line 251 and 258 the information related with PCR conditions (primers and PCR conditions).

Response 31: We have supplemented the information. The revised is The coding region of VvMYBC2L2 without the termination codon was amplified using the PrimerSTAR GXL DNA Polymerase (Takara, Japan) with the forward primer 5'- GGTCTAGAATGGTTGCAATGAGGAAGCCTGCAG -3' and reverse primer 5'- GGGGTACCTCCAAAAAGAAGTAGAGTTGTAAAG -3'. Reaction was the following thermal profile: 2min at 94 oC, 30 cycles of 30 s at 94 oC, 30 s at 56 oC, 1 min at 72 oC, then 10 min at 72 oC. The resulting fragments were digested with XbaI and KpnI and then cloned into the vector pBI221-GFP to generate pBI221-VvMYBC2L2-GFP constructs.” See line310.

Reviewer 2 Report

The authors present an interesting manuscript on the regulation of the gene expression in the flavonoids pathway in grapevine. The manuscript is well written and documented, the current knowledge in the field lay out and discussed well, and the methods are clearly stated and adequate. The results are interesting and add some valuable information to the knowledge of gene expression in this important crop plant.  I’m pleased to recommend the manuscript for publication with “Molecules”.

Still, the manuscript could benefit from some minor improvements and edits:

1)      The English has to be improved, and several grammar mistakes have to be corrected.  For example line 34: ‘The MYP family is one of the largest transcription factor…’; line 48: ‘ In strawberry, FaMYB1 could repress anthocyanin synthesis when was expressed in tobacco.’  Line 231: ‘Vitis vinifera cv Yatomi Rose was grew in the germ…’  Several more examples can be found throughout the manuscript. Typos are also rather common, e.g. line 39: ‘Nicotian tabacum..’; line 238: ‘Isolation of genmic DNA…’

2)      Scientific names should be used for all plants mentioned in the manuscript, e.g. line 47: lettuce plant; line 49: Chinese narcissus.

3)      It should be noted if the grapevine variety ‘Yatomi Rose’ is a red or white grape variety. The variety is not used in the traditional wine producing countries. Was there a reason to use this variety instead of some more widespread and economically important variety? What is Yatomi Rose used for? Wine production, table grape or rains?

4)      The meaning of R2R3-MYB transcription factor is not mentioned in the manuscript, please add.

Author Response

Dear editor and reviewers:

Thanks for your advice. According to the reviewer’ comments, we revised the manuscript and explain point-by-point the details of the revisions. Meanwhile, our manuscript has been edited by the International Science Editing ( http://www.internationalscienceediting.com ).

If you have any questions, please contact us. We look forward to hearing from you soon.

The following is our responses to the reviewer’s comments.

Point 1: The English has to be improved, and several grammar mistakes have to be corrected.  For example line 34: ‘The MYP family is one of the largest transcription factor…’; line 48: ‘ In strawberry, FaMYB1 could repress anthocyanin synthesis when was expressed in tobacco.’  Line 231: ‘Vitis vinifera cv Yatomi Rose was grew in the germ…’  Several more examples can be found throughout the manuscript. Typos are also rather common, e.g. line 39: ‘Nicotian tabacum..’; line 238: ‘Isolation of genmic DNA…’

Response 1: we have corrected these grammar mistakes and our manuscript has been checked using a professional English editing service. The revised is following:

 ‘The MYB family is one of the largest transcription factors families…’ see line 39.

‘ In strawberry, FaMYB1 could repress anthocyanin synthesis when it was expressed in tobacco.’,  see line 58.

Vitis vinifera cv Yatomi Rose was grown in the germ…’ see line 275.  

 ‘Nicotiana tabacum..’, see line 285.

 ‘Isolation of genomic DNA…’ see line 287.

Point 2: Scientific names should be used for all plants mentioned in the manuscript, e.g. line 47: lettuce plant; line 49: Chinese narcissus.

Response 2: We have revised. “while in Chinese narcissus (Narcissus tazetta)”; See line 59.  “In Arabidopsis, AtMYBL2 and AtMYB60 inhibit anthocyanin biosynthesis in Arabidopsis or Lactuca sativa plants” See line 57.

Point 3: It should be noted if the grapevine variety ‘Yatomi Rose’ is a red or white grape variety. The variety is not used in the traditional wine producing countries. Was there a reason to use this variety instead of some more widespread and economically important variety? What is Yatomi Rose used for? Wine production, table grape or rains?

Response 3: Sorrywe did not described the grapevine variety ‘Yatomi Rose’ clearly. V. vinifera cv ‘Yatomi Rose’ is a Red grape cultivar for table grape which were grown widespread in the north of China. The revised is “Vitis vinifera cv Yatomi Rose (Red table grape cultivar) was grown in the grape germplasm resource orchard of Shandong Institute of Pomology, Taian, Shandong, China.” See line 275.

Point 4: The meaning of R2R3-MYB transcription factor is not mentioned in the manuscript, please add.

Response 4: Sorry, we did not mention the meaning of R2R3-MYB transcription factor. The revised is “Based on the number of the highly repeat conserved domains (R), MYB proteins can be classified into four major types: 2R-MYB (R2R3-MYB), 3R-MYB (R1R2R3-MYB), 4R-MYB (R1R2R2R1), and MYB- related proteins (or 1R-MYB) [4, 5]. Among these MYB proteins, many R2R3-MYB proteins have been shown to be involved in the regulation of proanthocyanin or anthocyanin biosynthesis as positive regulators.” See line 40.

References:

4.      Dubos, C.; Stracke, R.; Grotewold, E.; Weisshaar, B.; Martin, C.; Lepiniec, L., MYB transcription factors in Arabidopsis. Trends Plant Sci 2010, 15, (10), 573-81.

5.      Martin, C.; Paz-Ares, J., MYB transcription factors in plants. Trends Genet 1997, 13, (2), 67-73.